# VARIANCE-REDUCED META-LEARNING VIA LAPLACE APPROXIMATION

## ABSTRACT

Meta-learning algorithms aim to learn a general prior over a set of related tasks to facilitate generalization to new, unseen tasks. This is achieved by estimating the optimal posterior using a finite set of support data. However, this estimation is subject to high variance due to the limited amount of support data for each task, which often leads to sub-optimal generalization performance. In this paper, we address the problem of variance reduction in gradient-based meta-learning and define a class of problems particularly prone to this. Specifically, we propose a novel approach that reduces the variance of the gradient estimate by weighing each support point individually by the variance of its posterior over the parameters. To estimate the posterior, we utilize the Laplace approximation, which allows us to express the variance in terms of the curvature of the loss landscape of our meta-learner. Experimental results demonstrate the effectiveness of the proposed method and highlights the importance of variance reduction in meta-learning.

## 1 INTRODUCTION

Meta-learning, also known as learning-to-learn, is a field concerned with the development of intelligent agents capable of adapting to changing conditions in the environment. The core idea of meta-learning is to learn a prior over a distribution of similar tasks, enabling fast adaptation to new environments given a few characterizing data points. This approach has proven successful in many domains, such as few-shot learning Snell et al. (2017), image completion Garnelo et al. (2018b), and imitation learning tasks Finn et al. (2017b).

One instantiation of this is Gradient-Based Meta-Learning (GBML), which was introduced in Finn et al. (2017a). GBML methods employ a bi-level optimization procedure, where the learner first adapts to a subset of data known as support data, and is then evaluated on a validation set called query data. The learned prior is then defined through the parameters of the adaptation procedure. A common issue for these methods is the uncertainty inherent in the adapted parameters given the support data, which can arise from measurement error, ambiguity between the tasks or limited support data. This variance can negatively impact the task-adaptation process, which essentially relies on the learner's ability to identify the task from the support data. This phenomena has also been referred to as meta-overfitting Finn et al. (2018); Yang & Kwok (2022); Wang et al. (2021); Charles & Konečný (2020).

To mitigate this issue, some methods propose to learn an aggregation procedure that directly models the distribution of the task-adapted parameters, Gordon et al. (2018); Kim et al. (2019). Using a learned network, these methods have the potential to be highly expressive, but suffer in the case of distribution-shift or label noise. In contrast, GBML methods are more robust and generally unbiased but have limited expressiveness when the support points can belong to different tasks.

In this work, we propose **L**aplace **A**pproximation for **V**ariance-reduced **A**daptation (**LAVA**), a method that introduces a novel strategy for aggregating the information of the support data in the adaptation process. The key idea of our method is to recognize each support point as inducing a unique posterior distribution over the task parameters. The posterior distribution can then be expressed as the joint posterior over the samples. Thus, our model is a form of Bayesian model-averaging, where each model is induced by a single point in the support. We approximate the posterior distribution w.r.t each support point through the *Laplace approximation*, which constructs a Gaussian distribution where the variance of the estimator is given by the Hessian of the negative log-likelihood. In turn, the joint

posterior is again Gaussian, from which the optimal value can be expressed as its mean. In contrast to other Bayesian GBML methods, the posterior approximation is not built on the full posterior but rather on the single posteriors induced by every point in the support data. This allows us to optimally aggregate the information these points share and reduce the variance of this estimate in the case of ambiguity.

Our contributions include an insight into the adaptation process of GBML methods by identifying a class of problems where GBML methods are particularly subject to high variance in their adaptation procedure. We introduce a method for modeling the variance that each data point carries over the parameters and optimally aggregate these posterior distributions into the task-adapted parameters. Finally, we demonstrate the performance of our method on synthetic and real-world experiments and showcase state-of-the-art performance compared to standard GBML methods.

## 2 PRELIMINARIES

In the meta-learning setting, we assume that there exists a distribution of tasks $p(\mathcal{T})$ that share a common generative process. Each task $\tau \in \mathcal{T}$ defines a space of inputs $X \subset \mathbb{R}^d$, a space of outputs $Y \subset \mathbb{R}^m$ and an unknown map between the two spaces $f_\tau : X \to Y$. Moreover, for each task, we are given a finite dataset sampled i.i.d., i.e. $D_\tau \subseteq X \times Y$. These points can be further divided into two sets, the support set $D_\tau^S$ and the query set $D_\tau^Q$ such that $D_\tau = D_\tau^S \cup D_\tau^Q$. The goal of meta-learning is to devise a learning algorithm that facilitates *few-shot learning* i.e. learning a task given only a limited amount of support data.

To enable this, we introduce the *meta-learner* $f_\theta$ which is trained to fast-adapt to the true function $f_\tau$. Meta-learning fast-adaptation consists of finding a parameterized inference process $\mathcal{A}_{\theta_0}$ that maps support data $D^S$ to a set of parameters $\theta_\tau$ such that $f_{\theta_\tau}$ minimize the empirical negative log-likelihood on the query dataset for each task:

$$\mathcal{L} = \frac{1}{T} \sum_{\tau=1}^{T} -\log p(D_\tau^Q | \mathcal{A}_{\theta_0}(D_\tau^S)). \tag{1}$$

Here, $\theta_0$ denotes the *meta-parameters* which are shared between all tasks and $T$ denotes the finite number of tasks used to compute the empirical loss. $p(D|\theta)$ denotes the likelihood of the data $D$ given model $f_\theta$.

### 2.1 GRADIENT-BASED META-LEARNING

In GBML, learning this adaptation process is formulated as a bi-level optimization procedure. Here, the meta-parameters $\theta_0$ are learned such that the inference process $\mathcal{A}_{\theta_0}$ corresponds to a single gradient ascent step on the likelihood computed using the support data and $\theta_0$. This leads to an approximate optimal set of parameters $\hat{\theta}_\tau$ for each task $\tau$:

$$\hat{\theta}_\tau = \theta_0 + \alpha \nabla_{\theta_0} \log p(D_\tau^S | \theta_0) \tag{2}$$

where $\alpha$ is a scalar value denoting the learning rate. Gradient-based Meta-Learning does not require additional parameters for the adaption procedure, as it operates in the same parameter space as the learner. Moreover, it is proven to be a universal function approximator Finn & Levine (2017), making it one of the most common models for meta-learning.

GBML can be formulated as a probabilistic inference problem from an empirical Bayes perspective Grant et al. (2018). The objective of GBML involves inferring a set of meta parameters $\theta_0$ that maximize the likelihood of the data $\mathbf{D} = \bigcup_\tau D_\tau$ for all tasks. Keeping the notation above and marginalizing over the task-adapted parameters, the GBML inference problem can be written as follows:

$$\theta_0 = \arg \max_\theta p(\mathbf{D}|\theta) = \arg \max_\theta \prod_\tau \int p(D_\tau^Q|\theta_\tau) p(\theta_\tau | D_\tau^S, \theta) d\theta_\tau \tag{3}$$

where $p(D_\tau^Q | \theta_\tau)$ corresponds to the likelihood of each task's data given the adapted parameters and $p(\theta_\tau | D_\tau^S, \theta)$ is the posterior probability of the task-adapted parameters.

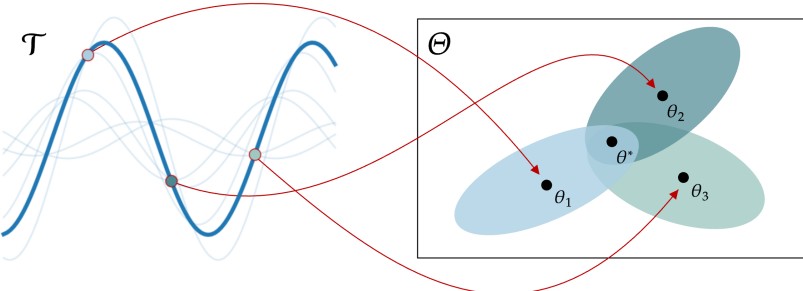

Figure 1: **On the left**: Three support points of a single task are marked out. These points are shared between different tasks which are marked out by the opaque curves. **On the right:** Each support point induces a distribution in the parameter space. The true function parameters $\theta^*$ lie at the *intersection* over the possible function values.

The integral in Equation (3) can be simplified by considering a *maximum a posteriori* (MAP) estimate of the posterior:

$$\theta_\tau^* = \arg\max_{\theta_\tau} p(\theta_\tau | D_\tau^S, \theta). \tag{4}$$

This simplifies the intractable integral to the following optimization problem:

$$\theta_0 = \arg\max_{\theta} \prod_\tau p(D_\tau^Q | \theta_\tau = \theta_\tau^*). \tag{5}$$

## 3 POSTERIOR ESTIMATION

The meta-learning objective can be reformulated with Equation 5 using as adapted parameters, $\theta_\tau$, the estimate from Equation 4. In GBML, the $\arg\max$ is approximated through a single gradient step on the loss function, as in Finn et al. (2017a). If the parameterized function $f_\theta$ is expressive enough and the tasks at hand are deterministic, we can make the following assumption:

**Assumption 1** (Solvable). *For each task $\tau$, there exists an optimal $\theta_\tau^*$ such that*

$$\mathbb{E}[\hat{\theta}_\tau] = \theta_\tau^*. \tag{6}$$

The assumption above implies that each task has at least one global minimum. The fact that it can be reached follows from the fact that the gradient is an unbiased estimator (see Appendix A.2) and that GBML is a universal function approximator Finn & Levine (2017). In practice, the expectation is approximated through the empirical mean and will be subject to estimation error following the variance of our estimate. In the following discussion, we fix a task and proceed by removing the subscript. Given that $D^S = \{x_i, y_i\}_{i=1}^N$, we can express the posterior estimate as:

$$\hat{\theta} = \frac{1}{N} \sum_{i=1}^N \theta_0 + \alpha \nabla_\theta \log p(y_i | \theta_0, x_i) = \frac{1}{N} \sum_{i=1}^N \hat{\theta}_i \tag{7}$$

where $\hat{\theta}_i$ denotes the task-adapted parameter for a single point $(x_i, y_i) \in D^S$. Assuming the support data is i.i.d., the variance of this estimator can be expressed as:

$$\text{Var}[\hat{\theta}] = \frac{1}{N^2} \sum_{i=1}^N \text{Var}[\hat{\theta}_i] \tag{8}$$

The variance of this estimate is dependent on the variance of the posterior estimate for each data point. In this paper, we argue that a source of this variance is that a single data point can belong to multiple tasks, which we denote as *task overlap*. An example of this can be seen by considering a sine wave regression problem. Let the distribution of tasks be sinusoidal waves with changing amplitude and phase, i.e., $y = A_\tau \sin(x + \phi_\tau)$. In this case, the space of tasks has dimension 2. A

single $(x, y)$ tuple can identify an infinite number of viable amplitudes and phases for which the sine would pass through that point, Figure 1 on the left. Thus, there exists a distribution of task-adapted parameters for which the inner loss function is minimized but only one task-adapted parameter when considering the intersection of solutions given by three points; see Figure 1 on the right. When using a single-point estimate, the information about this distribution of possible optimal task parameters is lost.

We can summarize the above in the following assumption. For ease of notation, let $f_\tau(x) = f(\tau, x)$ denote the true data-generating process.

**Assumption 2** (Task Overlap). *We can define task overlap as the condition that $\forall x \in \mathcal{X}$ the map $f(\cdot, x) : \mathcal{T} \to \mathcal{Y}$ is non-injective.*

An alternative formulation is that the conditional task-distribution $p(\tau|x, y)$ will have a non-zero covariance for all $(x, y) \in \mathcal{X} \times \mathcal{Y}$.

## 4 METHOD

In this section, we give an account of the source of the variance of the task-adapted parameters and propose a method to model the posterior more accurately. Given a finite set of support points $D^S$, we want to approximate the optimal posterior $p(\theta|D^S)$. Let $p(\theta|x_i, y_i)$ denote the posterior w.r.t to one data point. Given the assumption of task overlap (Assumption 2), this induces a distribution in parameter space. Another way of interpreting this posterior is that each data point provides evidence over what the possible true function could be. The true $p(\theta|D^S)$ will thus lie at the *intersection* over all marginal posteriors. Given the i.i.d. assumption, this implies that:

$$p\left(\theta|D^S\right) \propto \prod_{i=1}^{N} p(\theta|x_i, y_i).$$ (9)

Since we seek the MAP estimate of this distribution, we can disregard all terms not involving $\theta$, and assume a uniform prior on $p(\theta)$. Let us assume that $p(\theta|x_i, y_i)$ is normally distributed around its MAP estimate with a variance of $\Sigma_i$, i.e. $p(\theta|x_i, y_i) = \mathcal{N}(\theta; \hat\theta_i, \Sigma_i)$ with $\hat\theta_i = \arg\max_\theta p(\theta|x_i, y_i)$. From the product of Gaussian densities, the posterior is again proportional to a Gaussian $\mathcal{N}(\mu, \Sigma)$ with

$$\mu = \left(\sum_{i=1}^{N} \Sigma_i^{-1}\right)^{-1} \sum_{i=1}^{N} \Sigma_i^{-1}\hat\theta_i \qquad \Sigma = \left(\sum_{i=1}^{N} \Sigma_i^{-1}\right)^{-1}.$$ (10)

Since the MAP estimate of a normal distribution corresponds to its mean, we find that $\hat\theta = \mu$. If we assume that each marginal posterior $p(\theta|x_i, y_i)$ carries the same covariance matrix $\Sigma$, then the MAP estimate reduces to the empirical mean. By letting $\hat\theta_i = \theta_0 + \alpha\nabla_\theta \log p(y_i|x_i, \theta_0)$, we recover the standard GBML update. Thus, one interpretation of GBML methods such as MAML Finn & Levine (2017) is that the posterior estimation assumes an equal covariance across all marginal posteriors, which in turn implies that the MAP estimate equals the average of the parameters. This follows from the fact that we perform *a single gradient step* to approximate $\arg\max_\theta p(\theta|D^S)$. Due to linearity of the gradient operator, $\arg\max_\theta p(\theta|D^S)$ corresponds to the average of $\{\arg\max_\theta p(\theta|x_i, y_i)\}_{i=1}^{N}$. However, with the assumption of the joint posterior in Equation (9), this is not the correct estimate when the posterior is proportional to a product of distinct Gaussians.

### 4.1 LAPLACE APPROXIMATION

By Assumption 2, each $p(\theta|x_i, y_i)$ has a covariance structure that is dependent on the point $(x_i, y_i)$. In other words, each $x_i, y_i$ implies a different subspace over possible functions. To estimate this covariance, we appeal to the *Laplace Approximation*. To this end, we approximate each $p(\theta|x_i, y_i)$ as a normal distribution centered at $\hat\theta_i = \arg\max_\theta p(\theta|x_i, y_i)$ with covariance $\Sigma_i = H_i^{-1}$ where $H_i$ denotes the Hessian of the negative log-likelihood evaluated at $\hat\theta_i$. Inserting this in the estimate of $\mu$ in Equation (10), we find our new MAP estimate as:

$$\hat\theta = \left(\sum_{i=1}^{N} H_i\right)^{-1} \sum_{i=1}^{N} H_i\hat\theta_i.$$ (11)

We thus reach a MAP estimate that is weighted by the variance of each individual support point.

It is important to note that referring to it as an "approximation" may not be entirely accurate, as the flexible nature of parametric neural network models, such as our model $f_\theta$, allows for the precise shaping of probability distributions to meet specific requirements. Consequently, it is possible to shape the posterior distribution $p(\theta|x, y)$ to be normal, in which case the Laplace approximation would become exact. A parallel approach could involve estimating covariance of posterior $p(\theta|x, y)$ through a learned network $h_\psi : \mathcal{X} \times \mathcal{Y} \to \mathbb{R}^{d \times d}$. This approach is more akin to model-based methods of meta-learning, while our approach, through the Laplace approximation, remains model-agnostic.

## 4.2 VARIANCE REDUCTION

Our estimate in Equation (11) is in fact the variance-reduced distribution of a weighted sum of Gaussians. Consider $N$ random variables $Z_1, \ldots, Z_N$ distributed according to our posterior distributions $p(\hat{\theta}_i, H_i^{-1})_{i=1}^N$. Let $Z = \sum_{i=1}^N W_i Z_i$. We can define the variance reduction problem as:

$$\min_W \quad \mathrm{Var}\left(\sum_{i=1}^N W_i Z_i\right) \quad \textit{subject to} \quad \sum_{i=1}^N W_i = I. \tag{12}$$

**Proposition 1.** *The solution to Equation (12) equals the variance of our estimate defined in Equation (11). Consequently, the variance reduced distribution of $Z$ equals the joint posterior defined in Equation (9).*

A proof is given in the Appendix (Section A.3). Consequently, compared to parameter averaging defined in Equation (7), we achieve a lower variance for any support set $D^S \sim p(\tau)$. Thus, on expectation over all tasks, we achieve a variance-reduced estimate.

## 4.3 COMPUTING THE HESSIAN

Computing the Hessian amounts to evaluating the second-order derivatives of the negative log-likelihood loss. This is exceptionally costly for over-parameterized neural networks. To minimize the computational burden, we consider performing adaptation in a subspace of the parameter space. We opt for the implementation of CAVIA Zintgraf et al. (2019), which performs adaptation over a low-dimensional latent space $\mathcal{Z}$. This effectively allows us to compute the Hessian while keeping the computational cost reasonable and independent of the dimensionality of the input space.

Computing second-order derivatives, however, is known to be numerically unstable for neural networks, Martens (2016). We found that a simple regularization considerably stabilizes the training. In fact, following Warton (2008), we take a weighted average between the computed Hessian and an identity matrix before aggregating the posteriors. For all of our experiments, we substitute each Hessian $H_i$ in Equation (11) with the following:

$$\tilde{H}_i = \frac{1}{1 + \epsilon}(H_i + \epsilon I) \tag{13}$$

where $\epsilon$ is a scalar value and $I$ is the identity matrix of the same dimensionality of $H_i$.

## 5 RELATED WORK

**Gradient-Based Meta-Learning** methods were first introduced with MAML Finn et al. (2017a) and then expanded into many variants. Among these, Meta-SGD Li et al. (2017) includes the learning rate as a meta parameter to modulate the adaptation process, Reptile Nichol & Schulman (2018) gets rid of the inner gradient and approximates the gradient descent step to the first-order. In CAVIA Zintgraf et al. (2019), the adaptation is performed over a set of conditioning parameters of the base learner rather than on the entire parameter space of the network. Other works instead make use of a meta-learned preconditioning matrix in various forms to improve the expressivity of the inner optimization step Lee & Choi (2018); Park & Oliva (2019); Flennerhag et al. (2019).

**Bayesian Meta-Learning** formulates meta-learning as learning a prior over model parameters. Most of the work in this direction is concerned with the approximation of the intractable integral resulting

from the marginalization of the task parameters (Equation 3). This has been attempted using a second-order Laplace approximation of the distribution Grant et al. (2018), variational methods Nguyen et al. (2020), and MCMC methods Yoon et al. (2018). While these Bayesian models can provide a better trade-off between the posterior distribution of the task-adapted parameters and the likelihood of the data Chen & Chen (2022), they require approximating the full posterior and marginalizing over it. LLAMA Grant et al. (2018) proposes to approximate the integral around the MAP estimate through the Laplace Approximation on integrals. Given one gradient step, the posterior estimate is still performed by averaging (Equation 7). As we have shown, the averaging fails to account for inter-dependencies between the support points arising from task overlap.

**Model-based Meta-Learning** relies on using another model for learning the adaptation. One such approach is HyperNetwork Ha et al. (2016), which learns a separate model to directly map the entire support data to the task-adapted parameters of the base learner. In Gordon et al. (2018), this is implemented using amortized inference, while in Kirchmeyer et al. (2022), the task-adapted parameters are context to the base learner. Alternatively, the HyperNetwork can be used to define a distribution of candidate functions using the few-shot adaptation data Garnelo et al. (2018b;a) and additionally extend it using an attention module Kim et al. (2019). Lastly, memory modules can be iteratively used to store information about similar seen tasks Santoro et al. (2016) or to define a new optimization process for task-adapted parameters Ravi & Larochelle (2017). All of these methods can potentially solve the aggregation of information problem implicitly as the support data are processed concurrently. However, the learned model is not model-agnostic and introduces additional parameters.

## 6 EXPERIMENTS

To begin with, we test the validity of using the Laplace approximation to compute the task-adapted parameters for a simple sine regression problem. Additionally, we show how LAVA exhibits a much lower variance in the posterior estimation with respect to standard GBML. In the Appendix A.2, we demonstrate the unbiasedness of GBML and test the noise robustness of GBML and LAVA against a model-based approach.

We evaluate our proposed model on polynomial regression and dynamical systems tasks of varying complexity in regard to the family of functions and dimensionality of the task space. We compare the results of our model against other GBML models. In particular, our baselines include Model-Agnostic Meta-Learning (MAML) Finn & Levine (2017), Meta-SGD Li et al. (2017), ANIL Raghu et al. (2019), CAVIA, Zintgraf et al. (2019) as a context-based GBML method, PLATIPUS Finn et al. (2018), LLAMA Grant et al. (2018), BMAML Yoon et al. (2018) as Bayesian versions of GBML, VFML Wang et al. (2021) as a variance-reduced meta-learning method, and MetaMix Chen et al. (2021) as a meta-data augmented method. Experimental details are given in the Appendix A.1.

### 6.1 SINE REGRESSION

For the first experiment, we consider a typical sine wave regression problem to test the validity of the proposed assumptions and claims. Here, tasks are defined to be sine functions with varying amplitude and phase, i.e. $y = A_\tau \sin(x + \phi_\tau)$. In particular, the task's parameters $A_\tau, \phi_\tau$ are sampled from a uniform distribution between $[0, 5]$ and $[0, 2\pi]$ respectively, while $x$ is defined to be $[-5, 5]$. We sample a number of task parameters and generate the corresponding data points for the training and testing data. We consider 10 support data points for the inner step and 1000 for the query to test the quality of the adaptation. We use 10000 pre-sampled tasks for both the training and test dataset.

Given a single $(x, y)$ tuple, there exists a 1-dimensional space of sine-waves that pass through that point. This makes the aggregation challenging and thus allows us to test the benefits of approximating this subspace with the Laplace Approximation.

**Laplace Approximation Assumption** The first ablation aims at testing the quality of a Laplace approximation in modeling the distribution of the task parameters given each single data point. In the sine experiment, the dimensionality of the true task parameters is 2, allowing us to visualize the learned parameter space. To this end, we train both LAVA and CAVIA on the sine-wave regression problem with a context vector of dimensions 2.

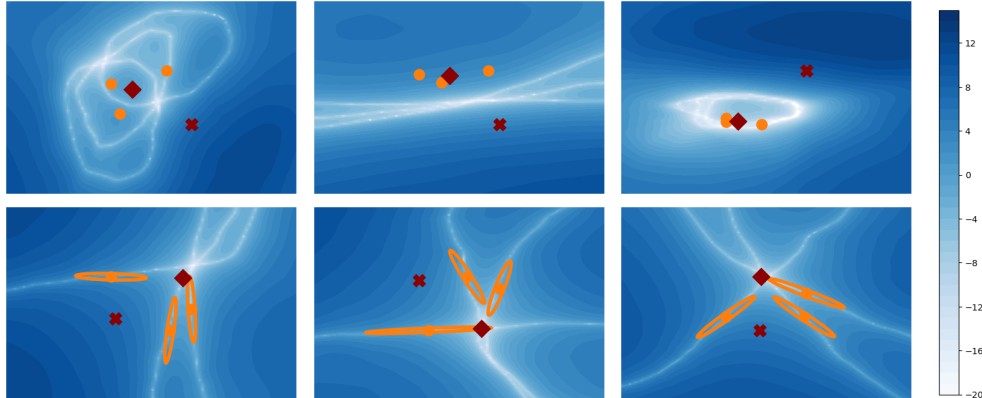

Figure 2: Sum of the logarithm of the loss for 3 support points over different parameters for the sine experiment. Values increase from white to dark blue. The red cross and the red diamond indicate the prior and the posterior, orange points are the single task-adapted parameters. **Top row:** Results for CAVIA. **Bottom row:** Results for LAVA, included is also the covariance for each support point.

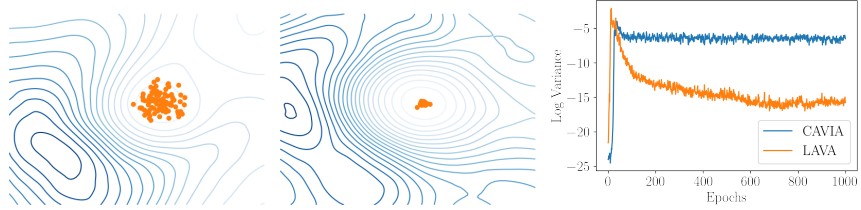

Figure 3: Variance of the task-adapted parameters given the same task but different support data points. **Left:** For CAVIA. **Center:** For LAVA. **Right:** Log variance of the distribution of the adapted parameters during training.

After the models have converged, we visualize the loss landscape induced by different support data points when using different task-adapted parameters. In particular, we sample a support dataset of 3 $(x, y)$ tuples and compute the logarithm of the mean squared error (MSE) between the prediction of both models and the true $y$ sampled from a grid of tasks. The idea is that each data point's loss is minimized by a continuous number of parameters. When summing up the losses of these support points, in fact, very well-defined valleys can be noticed in the loss landscape. Figure 2 shows the distribution of this sum of losses for a grid of task-adapted parameters as a heat map for CAVIA (top row) and LAVA (bottom row). Additionally, the prior context (red cross), the single task adapted parameters (orange dots) and the aggregated final posterior (red diamond) are also shown. For our method, we provide a visualization of the Hessian matrix for each single task-adapted parameter.

**Variance Estimation** For the second ablation, we evaluate the variance of the posterior estimate for both CAVIA and LAVA using the sine regression framework. The variance of the estimator describes the difference between the task-adapted parameters from the sampling of different support sets from the same task. For this experiment, we fix the sine task and sample 100 different support datasets of 10 $(x, y)$ tuples. For each of these support points, we compute the resulting task-adapted parameters. Figures 3 left and center show the spread of these distributions for CAVIA and our method, respectively. Note that the scale of the parameter space as well as the inner learning rate is the same for the two methods. Figure 3 on the right shows the log variance of the task-adapted parameters during training. This is computed by evaluating the variance of the estimated posterior from 100 different support datasets of the same task. As can be seen, at the beginning of training, the variance increases suddenly as the models learn to use these parameters to solve the meta-learning problem. However, while CAVIA's context parameters variance remains high during the rest of the training, our method learns to reduce it consistently.

| Models | $n = 2$ | $n = 4$ | $n = 6$ | $n = 8$ |
|--------|---------|---------|---------|---------|
| MAML | $0.85 \pm 0.15$ | $1.77 \pm 0.22$ | $2.53 \pm 0.30$ | $3.23 \pm 0.45$ |
| Meta-SGD | $1.35 \pm 0.21$ | $2.22 \pm 0.28$ | $2.82 \pm 0.42$ | $3.44 \pm 0.42$ |
| ANIL | $1.84 \pm 0.25$ | $2.86 \pm 0.33$ | $3.69 \pm 0.48$ | $4.44 \pm 0.49$ |
| PLATIPUS | $\underline{0.46 \pm 0.12}$ | $1.66 \pm 0.35$ | $2.53 \pm 0.44$ | $3.25 \pm 0.67$ |
| CAVIA | $0.87 \pm 0.14$ | $1.79 \pm 0.23$ | $2.55 \pm 0.26$ | $3.16 \pm 0.41$ |
| LLAMA | $0.86 \pm 0.14$ | $1.79 \pm 0.24$ | $2.57 \pm 0.29$ | $3.56 \pm 0.43$ |
| BMAML | $0.58 \pm 0.21$ | $\underline{1.14 \pm 0.17}$ | $\underline{1.57 \pm 0.21}$ | $\underline{2.07 \pm 0.34}$ |
| VFML | $1.38 \pm 0.22$ | $2.26 \pm 0.25$ | $2.93 \pm 0.39$ | $3.54 \pm 0.41$ |
| MetaMix | $0.86 \pm 0.09$ | $1.41 \pm 0.15$ | $1.74 \pm 0.19$ | $2.10 \pm 0.26$ |
| **LAVA** | $\mathbf{0.08 \pm 0.05}$ | $\mathbf{0.38 \pm 0.12}$ | $\mathbf{0.72 \pm 0.18}$ | $\mathbf{1.06 \pm 0.32}$ |

Table 1: MSE on Polynomial Regression with different degrees. Best results marked in **bold**, second underlined.

## 6.2 POLYNOMIAL REGRESSION

As a second set of experiments, we test if scaling the number of dimensions of the task distribution affects the relative performances of our method against a number of baselines. Here, the task distribution is represented by points sampled from polynomials of varying degrees with different parameters, i.e., $y = \sum_{i=0}^{n} a_i x_i^n$. By changing the degree of the polynomials, we can efficiently control the dimensionality of the task distribution and, thus, the complexity of the meta-problem. In particular, for this experiment, we use $n = \{2, 4, 6, 8\}$. Moreover, $x \in [-5, 5]$ and every $a_i \in [-1, 1]$. We use 10 support points for the inner adaptation step and 1000 query points for the outer loss and use a total of 1000 pre-sampled training tasks. Regarding the dimensionality of the context, we use a vector of the same dimensions of the task distributions, i.e., $\dim(\theta) = n + 1$. The loss function used is the MSE between the predicted $y$ and the true one for each task. Table 1 shows the final MSE mean and standard deviation for five different seeds. Our model clearly outperforms every other method independently on the dimensionality of the task distribution.

## 6.3 DIFFERENTIAL EQUATIONS

To further test more complex regression problems, we experiment with three different types of systems of ordinary differential equations and measure our performance on a held-out test dataset. We consider a mass-spring system from which we vary the mass $m$, a double pendulum where we vary the mass for each pendulum respectively and finally a multi mass-spring models in which we vary the spring constants between $n$ particles. Further details are given in the Appendix.

Each differential equation gives a different account of dimensionality and complexity. For the mass-spring, we predict the 1D position of the object. For the double pendulum, we predict the 2D position of the tip of the outer pendulum. This system also exhibits chaotic behavior and thus presents a more complex dynamics prediction problem. For the multi-mass spring, we predict the position of one particle. For each ODE, we fix the same initial conditions and vary only its parameters. The tasks are then defined by taking as input time $t$ and predicting the corresponding state at that time. We present our results in Table 2. For each ODE, we showcase superior performance compared to the baselines.

## 6.4 OMNIPUSH

In our final experiment, we showcase the performance of our model on a real-world dynamic prediction task. To this end, we utilize the *Omnipush* dataset Bauza et al. (2019) which consists of noisy, real-world data of different object dynamics when pushed by a robotic arm. The dataset consists of 250 different objects with 250 pushes per object. Each example consists of the pusher's relative position to the object and an angle denoting the direction of the push. The target is the change of the global pose of the object, $(\Delta x, \Delta y, \Delta \theta)$. The objects vary in shape and weight, which defines the task distribution. We report the mean squared error and standard deviation on forward dynamics prediction for support sizes $[10, 20]$ in Table 2. We use the same architecture as the other experiments. In this challenging dataset, a greater number of support points is required to accurately learn the task.

| Models | Mass-Spring | Pendulum | Multi Mass-Spring | Omnipush | |
| | | | | $|D^S| = 10$ | $|D^S| = 20$ |
| --- | --- | --- | --- | --- | --- |
| MAML | $1.53 \pm 0.12$ | $0.57 \pm 0.04$ | $3.87 \pm 0.23$ | $10.62 \pm 0.22$ | $9.59 \pm 0.22$ |
| Meta-SGD | $1.20 \pm 0.08$ | $0.33 \pm 0.05$ | $6.66 \pm 0.34$ | $\mathbf{9.03 \pm 0.32}$ | $\mathbf{7.80 \pm 0.24}$ |
| ANIL | $2.35 \pm 0.09$ | $0.84 \pm 0.05$ | $8.53 \pm 0.48$ | $10.36 \pm 0.28$ | $9.54 \pm 0.20$ |
| PLATIPUS | $10.3 \pm 38.0$ | $0.89 \pm 0.20$ | $4.17 \pm 0.47$ | $11.78 \pm 0.66$ | $9.77 \pm 0.56$ |
| CAVIA | $1.54 \pm 0.24$ | $0.46 \pm 0.03$ | $3.93 \pm 0.24$ | $9.59 \pm 0.30$ | $8.60 \pm 0.25$ |
| LLAMA | $2.01 \pm 0.15$ | $0.48 \pm 0.03$ | $4.20 \pm 0.18$ | $9.56 \pm 0.35$ | $8.40 \pm 0.35$ |
| BMAML | $\underline{0.73 \pm 0.09}$ | $\underline{0.29 \pm 0.04}$ | $4.71 \pm 0.33$ | $9.92 \pm 0.35$ | $9.02 \pm 0.25$ |
| VFML | $1.66 \pm 0.28$ | $0.49 \pm 0.04$ | $7.44 \pm 0.45$ | $9.53 \pm 0.35$ | $8.43 \pm 0.31$ |
| MetaMix | $1.26 \pm 0.05$ | $3.75 \pm 0.38$ | $7.05 \pm 0.50$ | $12.25 \pm 1.60$ | $12.09 \pm 1.31$ |
| **LAVA** | $\mathbf{0.55 \pm 0.10}$ | $\mathbf{0.18 \pm 0.01}$ | $\mathbf{2.74 \pm 0.19}$ | $\underline{9.19 \pm 0.31}$ | $\underline{7.87 \pm 0.31}$ |

Table 2: MSE $\times 10^{-2}$ on differential equations of varying complexity and Omnipush with different support sizes. Best results marked in **bold**, second underlined.

We achieve competitive performances compared to Meta-SGD, which learns a common update step across all tasks. In noisy settings with limited data, Meta-SGD could provide a more regularized model but also less expressive, as seen in the other experiments.

# 7 DISCUSSION AND CONCLUSION

In this paper, we characterized the problem of *task overlap* for under-determined inference frameworks. We have shown how this is a cause of high variance in the posterior parameters estimate for GBML models. In this regard, we have proposed LAVA, a novel method to address this issue that generalizes the original formulation of the adaptation step in GBML. In particular, the task-adapted parameters are reformulated as the average of the gradient step of each single support point weighted by the inverse of the Hessian of the negative log-likelihood. This formulation follows from the Laplace approximation of every single posterior given by each support data point, resulting in the posterior being the mean of a product of Gaussian distributions. Empirically we have shown how our proposed adaptation process suffers from a much lower variance and overall increased performance for a number of experiments.

The assumption of task overlap is sensible in the regression setting we have presented. We further consider a few-shot classification experiment on *mini-Imagenet* (see Appendix B.3) but achieve only comparable results to the baselines. Standard classification benchmarks represent adaptation problems that are inherently discrete and do not suffer from the problem of task overlap to the same extent as regression-like problems. Nevertheless, the discrete nature of classification problems presents an avenue for future work in the possibility of incorporating adaptation over a categorical distribution of parameters. A second limitation of our model is the computational burden of computing the Hessian. This may constrain the size of the context used, and consequently hinder the performance over more complex task distributions, in contrast to methods such as MAML or Meta-SGD where adaptation is done over the complete set of parameters. We compare our method to taking multiple steps in the inner loop and give an overview of the tradeoff between performance and compute in Appendix B.4. We show that the time complexity of our model is comparable to 5 inner steps of CAVIA, while showcasing superior performance.

An interesting extension to the proposed method would be to explore techniques to approximate the Hessian to speed up the computational time. This could be the Fisher information matrix or the Kronecker-factored approximate curvature Martens & Grosse (2015) to estimate the covariance of the Laplace approximation. Alternatively, it might be interesting to explore the direction of fully learning this covariance by following an approach similar to model-based methods.

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

# A  APPENDIX

## A.1  EXPERIMENTAL DETAILS

For all of the experiments and all the baselines, we fix the architecture of the meta-learner $f_\theta$ to be a multi-layer perceptron with 3 hidden layers of 64 hidden units together with ReLU activations. We use a meta batch size of 10 tasks and train all the models with Adam Kingma & Ba (2014) optimizer with a learning rate of $10^{-3}$. We use the inner learning rate $\alpha = 0.1$ for the adaptation step and MSE as the adaptation loss. All experiments were run for 5 different seeds to compute mean and standard deviations. For LLAMA we use $\eta = 10^{-6}$, for PLATIPUS we scale the KL loss by 0.1, for BMAML we use 10 particles and use MSE rather than the chaser loss for a fair comparison. Other experiment-specific details include:

- **Sine and Polynomial** regression: models are trained for $10^5$ outer loss steps, and the dimension of the context used is equal to the task's dimensionality.

- **Mass-Spring**: models are trained for $10^6$ outer loss steps, the dimension of the context used is 6, the ODE dynamics are defined as:

$$m\frac{d^2y}{dt^2} + b\frac{dy}{dt} + ky = 1 \tag{14}$$

  where $k, m, b$ are sampled uniformly in $[0.5, 2.0]$.

- **Double Pendulum**: models are trained for $10^6$ outer loss steps, the dimension of the context used is 4. The dynamics are described through the canonical Hamiltonian equations as:

$$\dot{\alpha}_1 = \frac{p_1 l_2 - p_2 l_1 \cos(\alpha_1 - \alpha_2)}{l_1^2 l_2 [m_1 + m_2 \sin^2(\alpha_1 - \alpha_2)]} \tag{15}$$

$$\dot{\alpha}_2 = \frac{p_2(m_1 + m_2)l_1 - p_1 m_2 l_2 \cos(\alpha_1 - \alpha_2)}{m_2 l_1 l_2^2 [m_1 + m_2 \sin^2(\alpha_1 - \alpha_2)]} \tag{16}$$

$$\dot{p}_1 = -(m_1 + m_2)gl_1 \sin\alpha_1 - A_1 + A_2 \tag{17}$$

$$\dot{p}_2 = -m_2 gl_2 \sin\alpha_2 + A_1 - A_2 \tag{18}$$

  with

$$A_1 = \frac{p_1 p_2 \sin(\alpha_1 - \alpha_2)}{l_1 l_2 [m_1 + m_2 \sin^2(\alpha_1 - \alpha_2)]} \tag{19}$$

$$A_2 = \frac{p_1^2 m_2 l_2^2 - 2p_1 p_2 m_2 l_1 l_2 \cos(\alpha_1 - \alpha_2) + p_2^2(m_1 + m_2)l_1^2}{2l_1^2 l_2^2 [m_1 + m_2 \sin^2(\alpha_1 - \alpha_2)]^2} \tag{20}$$

  for angles $\alpha_1, \alpha_2$ between the pendulums and generalized momenta $p_1, p_2$. From this, we find the coordinate $(x_2, y_2)$ of tip of the pendulum as

$$x_2 = l_1 \sin(\alpha_1) + l_2 \sin(\alpha_2) \tag{21}$$

$$y_2 = -l_1 \cos(\alpha_1) - l_2 \cos(\alpha_2). \tag{22}$$

  We sample parameters as $m_1, m_2 \sim \mathcal{U}[0.5, 1.5]^2$ and $g \sim \mathcal{U}[5, 15]$. The pendulum lengths $l_1, l_2$ are both constants with a value of 2.0.

- **Multi Mass-Spring**: models are trained for $10^6$ outer loss steps, the dimension of the context used equals 8. The dynamics can be simulated by iterating through each of the particles and summing the forces applied by the other particles:

$$v_i = v_i + \Delta t \frac{F_i}{m} \tag{23}$$

$$x_i = x_i + \Delta t v_i \tag{24}$$

  where $v_i, x_i$ denotes the velocity and position of particle $i$ respectively and $m$ denotes the mass. $F_i$ denotes the total spring force acted upon particle $i$:

$$F_i = \sum_{j=1}^{n} F_{ij} \tag{25}$$

$$F_{ij} = K_{ij}(1 - \|x_i - x_j\|)\frac{x_i - x_j}{\|x_i - x_j\|}. \tag{26}$$

---

**Algorithm 1** LAVA Pseudo-Code

---

**Require:** $p(\mathcal{T})$ distribution of tasks
**Require:** $\alpha, \eta, \epsilon$ hyperparameters.
**Ensure:** Output results

1: Randomly initialize $\theta_0$
2: **while** not done **do**
3:     Sample batch of tasks $\mathcal{T} \sim p(\mathcal{T})$
4:     **for all** $\tau \in \mathcal{T}$ **do**
5:         Sample $D_\tau^S, D_\tau^Q \sim \tau$
6:         **for all** $(x_i, y_i) \in D_\tau^S$ **do**
7:             Evaluate $\hat{\theta}_i = \theta_0 - \alpha \nabla_\theta \mathcal{L}(\theta_0, x_i, y_i)$
8:             Evaluate $H_i = \frac{d^2}{d\theta^2} \mathcal{L}(\hat{\theta}_i, x_i, y_i)$
9:             Evaluate $\tilde{H}_i = \frac{1}{1+\epsilon}(H_i + \epsilon I)$
10:         **end for**
11:         Evaluate $\tilde{H} = \sum_i \tilde{H}_i$
12:         Evaluate $\hat{\theta}_\tau = \tilde{H}^{-1} \sum_i \tilde{H}_i \hat{\theta}_i$
13:     **end for**
14:     Update $\theta_0 = \theta_0 - \eta \nabla_{\theta_0} \sum_{\tau \in \mathcal{T}} \mathcal{L}(\hat{\theta}_\tau, D_\tau^Q)$ using each $D_\tau^Q$
15: **end while**

---

where $F_{ij}$ denotes the force acted upoin particle $i$ by particle $j$. We sample spring constant $K_{ij} = K_{ji}$ between particle $i, j$ uniformly in $[0.5, 2.0]$. We let mass $m = 1$ and use a total of 4 particles.

- **Omnipush**: models are trained for $2 \times 10^5$ outer loss steps, and the dimension of the context used is $8$.

### A.2 GRADIENT-BASED META-LEARNING IS AN UNBIASED ESTIMATOR

Here, we show that GBML is an unbiased estimator. Define the loss for one task as:

$$\mathcal{L}(\theta, \tau) = \mathop{\mathbb{E}}_{x \sim p(x|\tau)} [\mathcal{L}(\theta, x)] \tag{27}$$

Then the gradient w.r.t $\theta$ is an unbiased estimator:

$$\mathop{\mathbb{E}}_{x \sim p(x|\tau)} [\nabla_\theta \mathcal{L}(\theta, x)] = \nabla_\theta \mathop{\mathbb{E}}_{x \sim p(x|\tau)} [\mathcal{L}(\theta, x)] = \nabla_\theta \mathcal{L}(\theta, \tau) \tag{28}$$

Moreover, we measure empirically the bias of GBML and LAVA estimators. As a comparison, we include a fully learned network implemented as a HyperNetwork Ha et al. (2016) that takes as input the entire support dataset and outputs the adapted parameters directly. Both the adaptation and the aggregation are learned end-to-end together with the downstream task.

We train these three models until convergence on the sine regression dataset. Then, we measure their performance on each task corrupted by Gaussian noise with a standard deviation of 3 on the support labels. The experiment is designed to test how the performance changes when increasing the support size. Figure 4 shows the difference in the loss between adaptation with and without noise for the three models and for different support sizes. Thus, we are effectively testing the ability of these estimators to recover the performances of the noiseless adaptation. Ideally, an unbiased estimator converges to the correct posterior with enough samples as long as the noise has zero mean. As can be seen in the figure, GBML methods are much more robust to these kinds of perturbations, while learned networks are not unbiased.

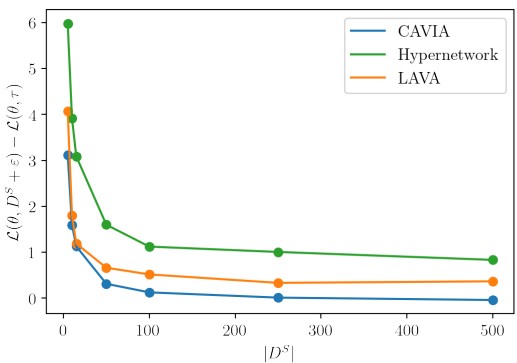

Figure 4: Scaled MSE when adding noise to the support labels for the sine experiment with increasing support size.

### A.3 VARIANCE REDUCTION

Below we give an account of the proof of Proposition 1. Consider the variance reduction problem defined in Equation 12

$$\min_{W} \quad \text{Var} \left( \sum_{i=1}^{n} W_i Z_i \right) \quad subject\ to \quad \sum_{i=1}^{n} W_i = I \tag{29}$$

We have that

$$\text{Var} \left( \sum_{i=1}^{n} W_i Z_i \right) = \sum_{i=1}^{n} W_i \Sigma_i W_i^T \tag{30}$$

By introducing a Lagrange multiplier $\lambda$, we reach the following optimization problem:

$$\min_{W} \quad F(Z, W) \tag{31}$$

$$F(Z, W) = \sum_{i=1}^{n} W_i \Sigma_i W_i^T + \lambda \left( \sum_{i=1}^{n} W_i - I \right) \tag{32}$$

By taking the derivative w.r.t $W_i$ and using the fact that $\Sigma_i$ is symmetric we find that

$$\frac{dF}{dW_i} = 2W_i \Sigma_i - \lambda \tag{33}$$

Setting this equal to $0$, we get

$$W_i = \frac{\lambda}{2} \Sigma_i^{-1} \tag{34}$$

From the condition defined in 29, we have

$$\frac{\lambda}{2} \sum_{i=1}^{n} \Sigma_i^{-1} = I \tag{35}$$

$$\lambda = 2 \left( \sum_{i=1}^{n} \Sigma_i^{-1} \right)^{-1} \tag{36}$$

$$\tag{37}$$

Plugging this into Equation 34, it follows that

$$W_i = \left( \sum_{i=1}^{n} \Sigma_i^{-1} \right)^{-1} \Sigma_i^{-1} \tag{38}$$

Given these weights, $W_i$, the distribution of $\sum_{i=1}^{n} W_i Z_i$ follows a normal distribution which is equivalent to Equation 10. $\square$

| Models | $|D^S| = 5$ | $|D^S| = 10$ | $|D^S| = 20$ |
|---|---|---|---|
| MAML | $0.834 \pm 0.129$ | $0.518 \pm 0.071$ | $0.316 \pm 0.042$ |
| meta-SGD | $0.63 \pm 0.08$ | $0.35 \pm 0.05$ | $0.20 \pm 0.02$ |
| ANIL | $0.72 \pm 0.09$ | $0.41 \pm 0.05$ | $0.23 \pm 0.03$ |
| PLATIPUS | $0.921 \pm 0.165$ | $0.386 \pm 0.064$ | $0.134 \pm 0.024$ |
| CAVIA | $0.651 \pm 0.105$ | $0.372 \pm 0.059$ | $0.199 \pm 0.029$ |
| LLAMA | $0.647 \pm 0.112$ | $0.371 \pm 0.058$ | $0.199 \pm 0.028$ |
| BMAML | $0.599 \pm 0.120$ | $0.349 \pm 0.075$ | $0.190 \pm 0.035$ |
| VFML | $0.64 \pm 0.08$ | $0.38 \pm 0.05$ | $0.21 \pm 0.02$ |
| MetaMix | $0.76 \pm 0.10$ | $0.67 \pm 0.09$ | $0.58 \pm 0.09$ |
| **LAVA** | $\mathbf{0.047 \pm 0.020}$ | $\mathbf{0.016 \pm 0.003}$ | $\mathbf{0.010 \pm 0.002}$ |

Table 3: Test MSE on sine regression with different support sizes.

## B    ADDITIONAL EXPERIMENTS

### B.1    ADDITIONAL SINE RESULTS

Here we provide additional results for the sine regression experiment. Using the same experimental settings described in Section 6.1 and Appendix A.1, we present MSE and standard deviations for 5 seeds for LAVA and baselines in Table 3. Additional qualitative results are shown in Figure 5

### B.2    ADDITIONAL COMPUTATIONAL TIME EXPERIMENTS

We present additional results on computational times for the polynomial experiment in Figure 6 for different number of degrees and size of support.

### B.3    MINI-IMAGENET

We further experiment with classification on the Mini-Imagenet dataset Vinyals et al. (2016). We use the training-set split as used in Ravi & Larochelle (2017) which leaves 64 classes for training, 16 for validation and 20 for test. We experiment with 5-way classification in either a 1-shot or 5-shot setting. We train the models for 1000 epochs and perform model selection by choosing the one with the best performance on the validation set. We present results on the test set in Table 4. In the 1-shot setting, we achieve results comparable to Meta-SGD and ANIL, outperforming Bayesian alternatives such as BMAML. In the 5-shot classification, we find that our model outperforms BMAML while being within the error bars of standard GBML methods such as CAVIA, MAML and Meta-SGD.

Standard classification benchmarks such as Mini-Imagenet test the capability of the model to incorporate high-dimensional data in the form of images. Some of the methods, such as the best-performing method BOIL, are optimized towards image data and attempt to efficiently learn a well-structured representation space of images, such that the adaptation reduces to modifying decision boundaries. In particular, few-shot image classification problems in this form are inherently discrete problems that do not suffer as extensively from the task overlap assumption as outlined in 2.

### B.4    A NOTE ON COMPUTATIONAL COMPLEXITY

A limitation of the described method lies in an increased time complexity. Computing the Hessian on each single support point can, in fact, severely affect the training time of methods that already require complex second-order calculations like in GBML. As a first consideration, we point out that the Hessian is computed on the contextual parameters only, leading to a more efficient computation rather than the full model parameters. When the network is expressive enough, this should lead to no difference in performance over the full adaptation framework.

Nevertheless, the Hessian computation can result in a sensible increase in computational time and LAVA is, in fact, more expensive than the standard GBML model. However, this computational complexity increase is paired with stronger performances. LAVA provides a more effective adaptation

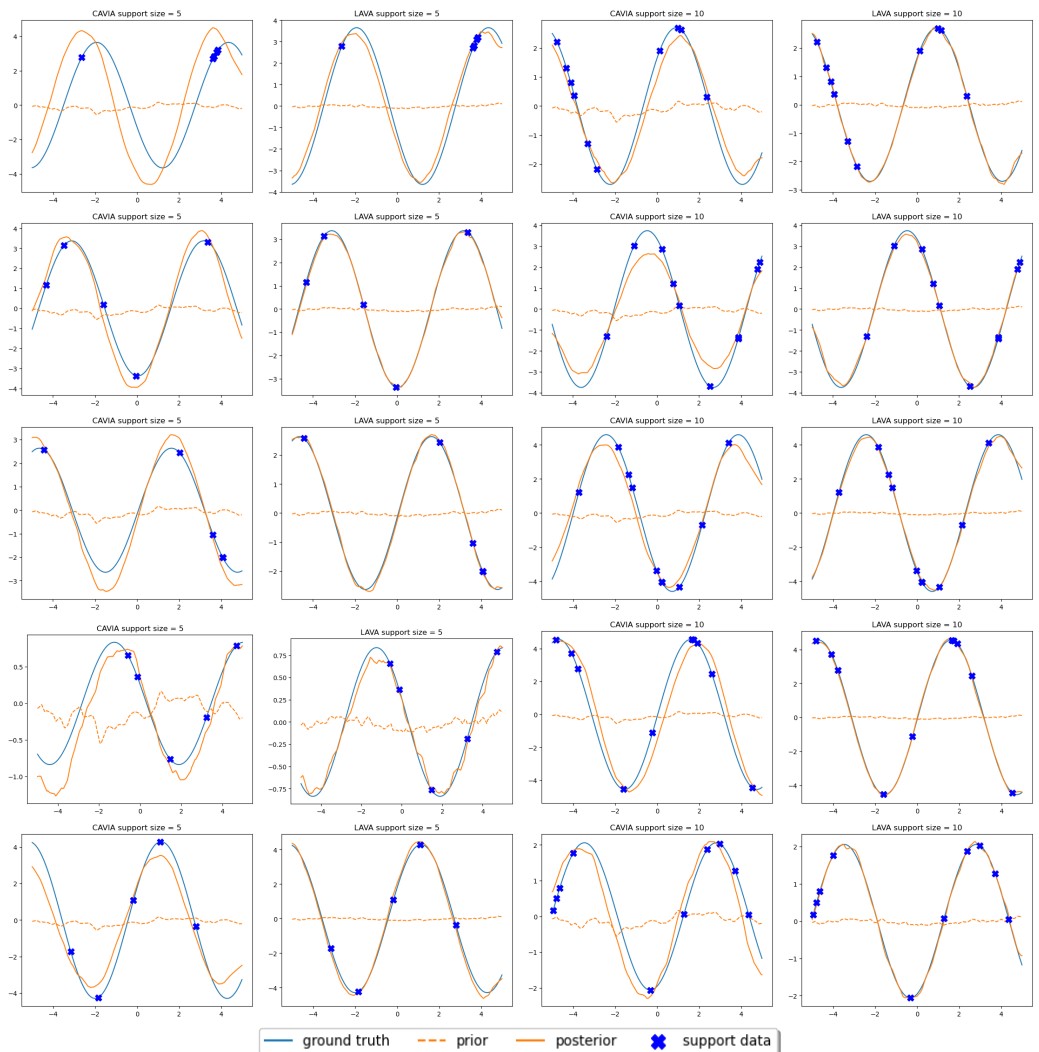

Figure 5: Qualitative results for the sine experiment for CAVIA and LAVA with 5 and 10 support size

| Model | 5-ways 1-shot | 5-ways 5-shot |
|---|---|---|
| ANIL | $45.94 \pm 0.94$ | $62.86 \pm 0.26$ |
| BMAML | $42.43 \pm 7.44$ | $59.76 \pm 0.43$ |
| BOIL | $\mathbf{50.12 \pm 0.33}$ | $\mathbf{64.72 \pm 0.40}$ |
| CAVIA | $47.84 \pm 0.41$ | $63.09 \pm 0.51$ |
| LLAMA | $40.19 \pm 0.85$ | $56.50 \pm 0.15$ |
| MAML | $\underline{48.60 \pm 0.80}$ | $\underline{63.19 \pm 1.57}$ |
| Meta-SGD | $46.18 \pm 0.45$ | $62.82 \pm 0.36$ |
| PLATIPUS | $34.71 \pm 0.68$ | $42.84 \pm 0.99$ |
| LAVA | $46.69 \pm 1.45$ | $61.51 \pm 0.97$ |

Table 4: Results Mini-Imagenet with support sizes 1 and 5

technique as one of its main features is the efficient use of the limited information given by the support. In this regard, LAVA provides a better trade-off between performances and computational complexity. To better analyze this trade-off we compared LAVA's performances against CAVIA in the Polynomial regression experiment by varying the number of inner loop adaptation steps. Figure 7

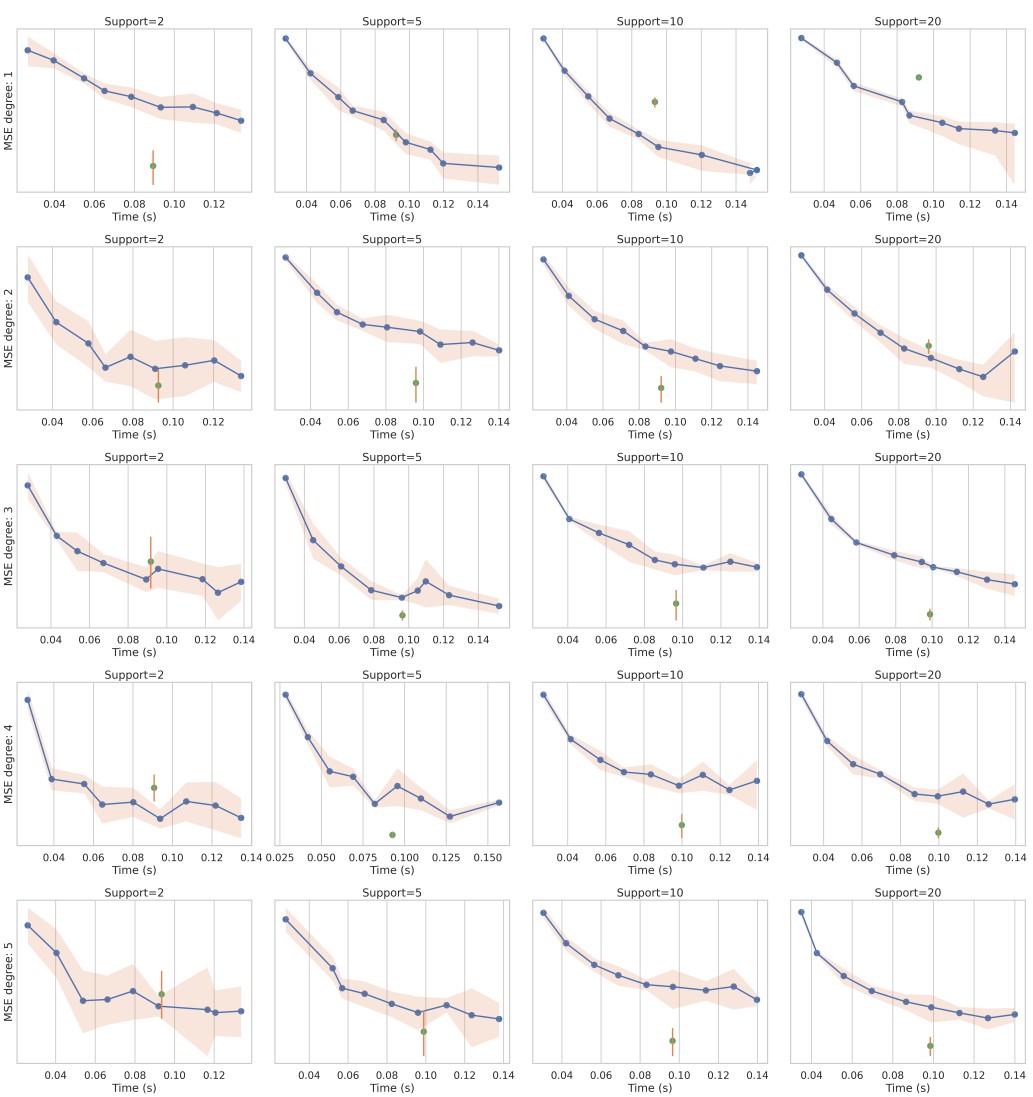

Figure 6: Computational times for the polynomial experiment of various degrees.

Figure 7: Computational times for third-degree polynomials with different sizes of support. The blue line represents results for CAVIA over a different number of adaptation steps. In comparison, LAVA achieves a computational time comparable to 6 inner-steps of CAVIA, while showcasing superior performance.

shows how LAVA has a computation complexity comparable to CAVIA with between 5 and 6 inner loop gradient steps. However, the performances of LAVA are consistently lower. A more extensive analysis is given in the Appendix in Figure 6.

There exists an inherent trade-off between computational complexity and model performance. Recent work exemplifies how deep learning modules benefit from more computational complexity, an example being the shift to Transformers from CNNs in computer vision tasks. LAVA represents another step in this direction as one of its main features is the efficient use of the limited information given by the support size at the expense of complex computations.

