# OpenReview forum: "Variance-Reduced Meta-Learning via Laplace Approximation"
_ICLR.cc/2024/Conference — Submitted to ICLR 2024_

### Official Review · Reviewer_idd4 · 2023-10-22

**Soundness:** 3 good
**Presentation:** 3 good
**Contribution:** 2 fair
**Rating:** 5
**Confidence:** 3

**Summary:**

This paper addresses the 'task overlap' problem, which enlarges the variance for posterior parameter estimate in GBML. A new approach, LAVA is proposed to alleviate this issue.

**Strengths:**

S1. This work identifies the task overlapping issue for GBML.

S2. To overcome the challenge of task overlapping, this work proposes a Hessian aided task adaption, and further employs Laplace approximation, variance reduction, as well as stabilized Hessian for efficiently solving it.

**Weaknesses:**

W1. Task overlapping needs more examples. Because this is the core to this work, it is beneficial to provide more real-world examples Current sinusoidal example appears to be oversimplified and not convincing enough.

W2. This work can benefit from additional experiments in real-world scenarios. It seems that on the only real-world dataset, omnioush, the proposed algorithm is not the most competitive.

**Questions:**

See weakness.

---

> ### Author Response · Authors · 2023-11-17
>
> We thank the reviewer for their comments and wish to address their concerns regarding real-world experiments below.
>
> We have included experiments on mini-ImageNet in the Appendix, and outline that real-world experiments such as classification fall outside the scope of the paper as they induce a discrete distribution of tasks. We agree with the reviewer that additional examples of task-overlap would help clarify the motivation of the paper.

---

### Official Review · Reviewer_ciAP · 2023-10-29

**Soundness:** 2 fair
**Presentation:** 2 fair
**Contribution:** 2 fair
**Rating:** 5
**Confidence:** 4

**Summary:**

The authors characterized task overlap for GBML, and proposed a solution to address high variance in GBML, which reduced the variance of the gradient estimate by weighing each support point individually by the variance of its posterior over the parameters.

**Strengths:**

The idea is of task-overlap is good, which is demonstrated in Figure 1.

The idea of aggregating of parameters is good and new.

The experiments on the toy example, sine function regression, contain good discussion about the variance reduction.

**Weaknesses:**

**The presentation can be further strengthened.**

* The notation keeps changing, which is very unclear. Please make notation clear and consistent.
* The type of reference, eg. in-text reference, needs to be checked.
* Eq 1. is not clear. It is better to rewrite the objective.

**Please reconsider the assumptions and Proposition 1.**
* Eq 3. is unclear, as $D_{\tau}^{Q}$ is just one realisation of $D_{\tau}$. It seems that the authors miss another expectation there given that the authors claim that the uncertainty comes from data points.
* Meanwhile, the authors claim that $D$ is a collection of all tasks, is it finite or countable infinite? Are there any uncertainty in the generating process of tasks themselves?
* In theory, it is even difficult to have a local solution of MAML's objective. To ensure it, we often need additional assumptions. The assumption 1 and the paragraph underneath do not make sense from a theoretical aspect though the authors try to provide some theory.
* The definition of $h_\psi$ on page 5 is problematic as covariance needs to be positive semi-definite.
* I do not understand why the authors need to have Proposition 1 if it is a well-known result.




**From the experiments in Appendix, it seems that the method does not work well on mini-ImageNet.**

**Questions:**

Why does the proposed method perform worse than MAML on mini-ImageNet?

---

> ### Author Response · Authors · 2023-11-17
>
> We thank the reviewer for their insightful comments and wish to address their concerns below.
>
> **The presentation can be further strengthened.**
>
> We agree with the reviewer that the presentation can be clarified. We have updated the notation and type of references in a revised version of the paper and hope the clarity has improved. We have additionally edited the objective described in Equation 1.
>
> **Equation 3 is unclear**
>
> The dataset is fixed with a finite collection of tasks and samples per task. In Equation 3, we define our objective as the maximum likelihood estimate of the parameters $\theta$. This is similarly expressed in Equation 2 in [1]. The uncertainty arises in the estimate of the posterior p(\theta_\tau|x,y,\theta) precisely because the number of samples for each task are finite. We introduce a better estimate of this that reduces this uncertainty.
>
> [1]: Erin Grant, Chelsea Finn, Sergey Levine, Trevor Darrell, and Thomas Griffiths. Recasting gradient-based meta-learning as hierarchical bayes. International Conference on Learning Representations (ICLR), 2018.
>
>
> **In theory, it is even difficult to have a local solution of MAML's objective. To ensure it, we often need additional assumptions. The assumption 1 and the paragraph underneath do not make sense from a theoretical aspect though the authors try to provide some theory.**
>
> We agree with the reviewer that MAMLs objective is a non-convex optimization problem and thus lacks a guarantee of convergence. Nevertheless, the effectiveness of gradient-based meta-learning has been proven in many scenarios thus we include the optimality as an assumption. We include this assumption to imply that there exists a local minimum for the task as this yields a non-zero Hessian. We agree with the reviewer that the assumption of global minimum
>
> **Definition of $h_\psi$.**
>
> The covariance does indeed have to be positive semi-definite. Enforcing such a condition is possible e.g. using the Cholesky decomposition similar to as defined in [2]. We have updated the definition of the map to be $h_\psi : \mathcal{X} \times \mathcal{Y} \to \mathbb{S}^{d}_+$ (the space of positive semi-definite matrices)-
>
> [2] Xu, Kailai, Daniel Z. Huang, and Eric Darve. "Learning constitutive relations using symmetric positive definite neural networks." Journal of Computational Physics 428 (2021): 110072.
>
>
> **I do not understand why the authors need to have Proposition 1 if it is a well-known result.**
>
> We include Proposition 1 to draw a connection between the MAP estimate of the task-adapted parameters and the problem of variance reduction. To the best of our knowledge, we have not found any direct references for this result, which motivates us to include it in our presentation.
>
>
> **Why does the proposed method perform worse than MAML on mini-ImageNet?**
> We wish to clarify our assumptions about the task distribution. An additional assumption we make is that the distribution of tasks p(\theta|x, y) is an unimodal and continuous distribution, as the induced map from task to parameters space is a continuous function. Image classification introduces a discrete, multi-modal distribution over the task-manifold, and as such is not suitable to be approximated with a normal distribution. This assumption will be clarified in Section 3 on posterior estimation.

---

> > ### Comment · Reviewer_ciAP · 2023-11-22
> >
> > I have read the authors' responses and decide to keep my score.

---

### Official Review · Reviewer_mLEP · 2023-10-29

**Soundness:** 3 good
**Presentation:** 2 fair
**Contribution:** 2 fair
**Rating:** 5
**Confidence:** 4

**Summary:**

The paper considers improving gradient-based meta-learning from the perspective of approximate bayesian inference. Here the authors show that vanilla MAML’s adaption can be seen as aggregating Gaussian-approximated posterior distribution over each support example with the same covariance matrix. Instead of using the same covariance, the authors apply Laplace approximation so that different support examples have its own covariance matrix (which is the Hessian of the example’s negalive log-likelihood). The proposed method LAVA then uses the Hessian-weighted individual-example-adapted parameters as the final adapted model parameters. The authors show that this new approach reduces the variance of the parameter estimates. Experimentally, results are shown over 3 few-shot regression tasks which demonstrate the performance advantage of LAVA over existing meta-learning methods.

**Strengths:**

- The idea of aggregating individual example’s posterior distribution in the context of meta-learning is to my knowledge novel.
- The connection between MAML (which the authors argue treats each example’s posterior covariance as equal) and LAVA (which allows Laplace approximation in individual example’s posterior) is novel and insightful.

**Weaknesses:**

Although I find the paper’s proposed idea interesting,  I find the choice of technical language, explanation, and the experimental results a bit lacking in its current form for publication:

- **Method is computationally too expensive**.
    - Despite the authors performing an analysis in Figure 6 and 7 showing the time cost of 1 inner step of LAVA is comparable to 6 inner steps of CAVIA, it is important to note that this relationship depends on (and should be linear to) the dimensionality of the adaptable parameter space of LAVA, as the Hessian computation takes $O(d)$ backprops where $d$ is the dimension of $\theta$.  If we choose the number of adaptable parameters to be larger (for example, 100 context parameters in the image classification task in the original CAVIA paper), the computation and time cost of LAVA would be much greater than CAVIA.
    - Given the worse computation requirement of LAVA compared to CAVIA and MAML, I believe it is necessary for the authors to further discuss and investigate possible techniques for computation savings. For example, instead of computing and storing the Hessian, it is conceivable to use techniques such as 1) only performing Hessian vector products (which only takes constant back props to compute) and 2) using conjugate gradients to approximately find the (inverse Hessian, vector) product. I believe these explorations are necessary for the scaling of the proposed method, as I don't believe purely scaling up the non-adaptable parameters while keeping the number of adaptable parameters to a very small number (e.g. $<10$) can provide sufficient performance in the case of LAVA on real world problems.
- **Evaluating on problems with higher inherent dimensions** The three few-shot experiments are all internally parameterized by a small number of parameters ($<10$), which might make it appropriate consider LAVA which could adapt a correspondingly small number of context parameters. However, for real world parameters, the inherent problem dimensions could be much higher or unknown. (One experiment setup that more closely captures this property is the image completion task used in CAVIA.) The current experiments in the paper haven't demonstrated the ability of LAVA to handle such problems.
- **Evaluating on another model architecture.** All the three few-shot experiments are performed using the same 3 layer 64 hidden unit MLP architecture. It is useful to also demonstrate that the proposed method can perform well for other architectures (different computation blocks, depths, and widths).
- **Comparing gradient-based meta-learning methods under the same number of inner-problem backprops used**. In the experiment section, the authors have reported performances of different baseline gradient-based meta-learning methods. However, it’s not clear to me whether all these methods are restricted to using 1 inner steps (the same as LAVA). If so, I want to highlight that LAVA’s single inner adaptation update step requires number of backprops proportional to the adaptable parameter dimension, while the other methods only need 1 backprop per single inner adaptation step. Thus, for fairness, the authors should also allow the other baseline methods to use the same number of backprops by allowing them to take more inner adaptation steps (preferably during meta-training, at least during meta-test). This might improve the performances of some of the baseline methods.
- **Unsatisfying explanation on the lack of performance on Mini-imagenet**. In Table 4 in the Appendix, the authors compare LAVA with other meta-learning methods on Mini-Imagenet (arguably the most commonly used few-shot meta-learning benchmark) and find the performance to be worse than many other methods. The authors give an explanation that the few-shot classification classification are inherently discrete problems that do not suffer as extensively from the task overlap. However, this claim seems incorrect to me, as observing a single image, label pair from a few-shot classification task is also insufficient to determine what the rest of the classes are in the task. Thus there is also task overlap according to the authors’ definition. Besides, it’s only unclear how the discreteness of the labels would play a role in this, as the learning parameters $\theta$ are in a continuous space.
- **choice of technical language**
    - **Assumption 1**. It is a bit unclear how to define the optimal $\theta_{\tau}^*$. To my understanding, the right-hand side of Equation (6) should not guarantee to be the best possible $\theta$ in the space $\Theta$ for the task $\tau$. Instead, it makes more sense use $\theta_{\tau}^*$ to denote the one-step gradient adapted model parameters over an infinite number of support examples sampled from the task $\tau$. (In fact this is what has been shown in Appendix A.2). However, if more than one gradient steps are taken, the unbiasedness (and thus Assumption 1) would no longer be true.
    - **Assumption 2**. The notation $f_\tau(x)$ is defined without exactly specifying what its output means until Assumption 2. Besides, in the interpretation of Assumption 2, the authors mention an alternative formulation is that the conditional task-distribution have a non-zero “covariance”. However, I believe this terminology should be changed to “having a non-singleton distribution support”.
- **Equation 9 is incorrect without additional assumptions**. We notice that the single-example posterior terms on the right hand side is each proportional to product of the example likelihood and the prior $\propto p(y_i \mid x, \theta) p(\theta)$. Hence the product on the right hand side of Equation 9 would be proportional to the product of $N$ prior probabilities $p(\theta)^N$. In contrast, the left hand side of Equation 9 is still proportional to only $p(\theta)$, making the prior weighted more heavily on the right hand side of (9). A way to fix this is to introduce the uniform prior assumption earlier than Equation (9) instead of after.
- **The explanation of the variance reduction is a bit lacking**. Equation (12) formulates a variance reduction problem. However, the current writing misses the connection to its use of Assumption (1) which would imply that the estimators $Z$ under different values of $\{W_i\}$ would have the same expectation. Without clearly specifying this, the variance reduction problem seems out of place and unmotivated.

**Questions:**

- On page 5, it was mentioned that “Consequently, it is possible to shape the posterior distribution $p(\theta | x, y)$ to be normal”. It’s not clear to me how this would be possible for general neural network predictive models. Can the authors further explain this?
- How many context parameters (adaptable inner loop parameters) are used in the MiniImagenet experiment? How long would the training take compared to other methods on this application?
- Why are the dotted prior curves sometimes different for the same column in Figure 5 in the Appendix?

---

> ### Author Response · Authors · 2023-11-17
>
> We thank the reviewer for a thorough and insightful review. We would like to address the questions and concerns raised below:
>
> **Method is computationally too expensive.**
>
> The Hessian computation indeed requires O(d) more computational time. However, this calculation can be done in parallel, which by assuming d^2 processes yields a Hessian computation that is only twice that of the Jacobian, trading computational time for memory complexity.
> We agree with the reviewer in that with a sensible increase in the dimensionality of the task space the computational cost would be infeasible. Most real-world problems have an intrinsic dimensionality low enough to allow these memory optimizations. In the mini-ImageNet experiments, we experiment with a context size of 100 within a reasonable computational time.
> As the reviewer points out, additional measures can be taken to reduce the time and memory complexity of the problem. This is an important avenue for future work and we will append Section 4.3 with an additional discussion on alternative methods to compute the Hessian.
>
>
> **Evaluating on problems with higher inherent dimensions and Evaluating on another model architecture.**
>
> While it is true that in the experiments described in the main text of the paper, both the model and the context size are relatively small, we do however provide results for the Mini-ImageNet experiments in the Appendix. For this experiment, the model used is a CNN architecture of similar size to the other baselines (4 CNN layers and 1 or 2 MLP layers) and a context size of 100 dimensions. LAVA does not achieve state-of-the-art performances on this experiment but it provides evidence for the usability of LAVA on more high-dimensional experiments.
>
> **Comparing gradient-based meta-learning methods under the same number of inner-problem backprops used.**
>
> Equating the models under the same computational complexity is an interesting proposition, however, we would like to point out some potential pitfalls: it introduces overfitting in the GBML methods, especially in higher dimensions, which may not yield an accurate comparison. Secondly, it neglects the tradeoff between computational and memory complexity that our method can exploit, by parallelizing the Hessian computation which has no direct equivalent in GBML methods, as introducing more gradient steps can only be done sequentially. We thank the reviewer for pointing out that a further analysis of the complexity of the method is warranted, and we will add a discussion on this in a future version of the manuscript.
>
>
> **Unsatisfying explanation on the lack of performance on Mini-imagenet**
> We agree with the reviewer that there is still task-overlap in the sense of Assumption 2 in the Mini-ImageNet dataset. We wish to clarify our assumptions about the task distribution. An additional assumption we make is that the distribution of tasks p(\theta|x, y) is an unimodal and continuous distribution, as the induced map from task to parameters space is a continuous function. Image classification introduces a discrete, multi-modal distribution over the task-manifold, and as such is not suitable to be approximated with a normal distribution. This assumption will be clarified in Section 3 on posterior estimation.
>
>
> **Choice of technical language**
>
> **Assumption 1:**
> The expectation on the left-hand side of Equation 6 is over different samples x from the support dataset. This implies that the number of support samples is finite, but the expectation is taken over different samples of a set of x. Assumption 1 then implies that each $\hat{\theta}_i$ is the same as the optimal on expectation. This justifies the variance reduction as each Z_i in Equation 12 is then assumed to have the same mean.
>
> **Assumption 2:**
> In the beginning of section 2, we define $f_\tau : X \to Y$ as a map between two spaces X, Y for a specific task $\tau$. Its output defines different tasks as $\tau$ is varied. We agree with the reviewer that a non-singleton distribution support is a more correct terminology, and will update the end of section 3 in a revised version of the manuscript.
>
> **Equation 9 is incorrect without additional assumptions:**
> We thank the reviewer for their suggestion. The prior $p(\theta)$ is indeed assumed to be uniform. We will clarify these assumptions in Section 2.1.
>
>
> **The explanation of the variance reduction is a bit lacking**
> The observation that the expectation of each Z_i is equal to the optimal as is stated in Assumption 1. That the Z_i have an equal mean is indeed an assumption in the variance reduction problem. We will add this assumption in proposition 1.
>
> **Why are the dotted prior curves sometimes different for the same column in Figure 5 in the Appendix?**
>
> It is true the prior should be the same within each model in Figure 5 in the appendix. The scale of the y-axis is however different between some of the plots making the prior seem different.

---

> > ### Comment · Reviewer_mLEP · 2023-11-22
> > **Response to author rebuttal**
> >
> > I acknowledge the authors’ rebuttal response. I have the following comments:
> >
> > - **Parallelizing the hessian computation**. The authors mention that the Hessian computation can be parallelized. This is indeed possible. However, I also want to point out the practical difficulties of doing this. To compute the Hessian, we need $d$ Hessian-vector products to fully instantiate the Hessian matrix. These Hessian-vector products can be parallelized, but each would already require double the amount of memory used by a regular backprop. Thus to truly calculate the Hessian in parallel, one would need a compute system whose total (GPU) memory size is $O(d)$, which would quickly become difficult to achieve when $d \ge 100$.
> > - __Performing adaptation on mini-imagenet.__ I think one additional approach the authors could consider in the framework of LAVA is to consider treating a group of examples as a mega-example. In miniimagenet 5 way 5 shot, I could imagine partitioning the training set into 5 subsets, where each subset has one unique example from each class. Then the authors can perform LAVA on the 5 model posteriors’ (each conditioned on one subset) weighted combination instead of on the combination of 25 individual example’s posterior. I think this might help get rid of the multi-modal distribution problem mentioned by the authors.
> >
> > Overall, taking the authors paper and rebuttal into consideration, I decide to maintain my current score. The primary reasons of my score not being higher are:
> >
> > 1. __The paper could benefit from some partial rewriting.__ For example, if stated clearly, assumption 1 should actually be a proposition or lemma, and the connection of Proposition 1 to variables in the paper (for example $Z_i$ corresponds to the adapted model parameter over a single example whose distribution follows a Gaussian) should be more clearly explained (it took me some time to fully understand it).
> >
> > 2. __Demonstrating LAVA’s competitiveness on more than one architecture and another task with unknown dimension.__ Even though the authors have shown results on mini-imagenet, it is generally a negative result that shows LAVA not performing as well as some of the baselines. Excluding this experiment, most of LAVA’s results are reported on a 3 layer 64 hidden unit MLP architecture. It is expected that the authors experiment with at least some different architectures of different widths and depths. Besides, I would like to suggest the author consider experimenting on the image completion task considered in CAVIA (as that task has a potentially higher and unknown hidden dimension).

---

### Official Review · Reviewer_gPmV · 2023-11-03

**Soundness:** 3 good
**Presentation:** 4 excellent
**Contribution:** 3 good
**Rating:** 6
**Confidence:** 3

**Summary:**

This paper presents a new gradient-based meta-learning algorithm that is able to decrease the variance of the learned parameters. Taking a Bayesian perspective, it weights each of the support data points by the variance of the posterior that it induces over the parameters, approximated using a Laplace Approximations. Experiments show improvements in meta-learning ODEs w/o substantially increasing computation time, but is inconclusive for a real-world physics simulation problem.

**Strengths:**

# Originality and significance #

As far as I know, this paper is the first application of the Laplace Approximation to GBML. The results are promising, so I think this work would be relevant to the community.

# Clarity #

The paper is written very clearly and is straightforward to understand. The authors explain some of the assumptions made.

# Quality #

The proposed algorithm is simple and straightforward, with approximate (but incomplete) mathematical backing.  A good diversity of problems and baselines are used in the experiments, and the experimental results are promising.

**Weaknesses:**

1. The experiments do not contain ablations in certain axes. The problem dimensionality is varied, but the model dimension (the network architecture) is not.
2. The assumptions of the algorithm is not clear. The variance calculation implicitly assumes that $\theta_0$ is deterministic, but is that reasonable given that it depends on previous parameter updates (which are not deterministic)?

**Questions:**

1. The standard errors for the Omnipush results are large compared to the differences in algorithms' performances. Have you tried increasing the number of trials to reduce it?
2. How do the results vary when the model architecture varies, e.g. when the number of parameters in the neural network increases?
3. Is there a proof of the claim made in the first paragraph of page 5?

---

> ### Author Response · Authors · 2023-11-17
>
> We thank the reviewer for their insightful comments and wish to address their concerns below.
>
> **The experiments do not contain ablations in certain axes. The problem dimensionality is varied, but the model dimension (the network architecture) is not.**
>
> We thank the reviewer for their suggestion and agree that a study of the network architecture would be a valuable addition to the paper. On page 5, paragraph 1, we make the assumption that our network is expressive enough to model the task distribution as a Gaussian in parameter space. This raises an interesting question of what architectural choices affect the expressivity in the parameter space. [1] Suggests that the number of layers is more important than the network width for gradient-based meta-learning. It would be interesting to extend this analysis to what architectures allow for arbitrary shaping of the parameter space. We will append this analysis to the experimental section in a revised version of the manuscript.
>
>
> [1] When MAML Can Adapt Fast and How to Assist When It Cannot, Arnold et. al. (2019)
>
>
> **The assumptions of the algorithm is not clear. The variance calculation implicitly assumes that $\theta_0$ is deterministic, but is that reasonable given that it depends on previous parameter updates (which are not deterministic)?**
>
> The reviewer is correct that the variance calculation only takes into account the variance of the sampled support data. The variance of the task-adapted parameters arises from both the finite sampling of the support and the distribution of $\theta_0$, which as the reviewer points out arises during stochastic gradient descent. In this work, we study the variance of the task-adapted parameters when $\theta_0$ is given and the support data is varied. In the paper, we show that GBML methods might still suffer from a high-variance estimation even when having access to the optimal $\theta_0$. We can make this assumption, and focus on the adaptation-induced variance only, as $\theta_0$ is not dependent on the current sampling of the support, which affects only the task-adapted parameters. We will further clarify this in Section 3.
>
>
>
> **Is there a proof of the claim made in the first paragraph of page 5?**
>
> The adaptation process essentially induces a continuous map $M : T \to \theta$ from which we model the image $M(T)$ as a Normal distribution using the Laplace approximation. For this to be possible, we assume there exists a diffeomorphism between the true task distribution and the normal. This might not be true in general, for example if the topology of the space of tasks $T$ is not simply-connected or doesn’t contain only one connected component.
>
> If these assumptions are satisfied, $f$ will find a solution given that it is a universal function approximator. This is reasonable to assume given $f$ is of a sufficient size. As the reviewer has suggested a study of the effect of parameter count and performance would be a valuable addition to validate this claim. We will append the experimental section with further results on different architectures and add it to a revised version of the manuscript.

---

> > ### Comment · Reviewer_gPmV · 2023-11-22
> > **Thanks to the authors for your response**
> >
> > After reading the other reviews and responses, I have decided to keep my score. The main reason that I do not give a higher score are the weaknesses in the experimental results, which some of the other reviewers have also mentioned. I would encourage the authors to conduct more experiments with different network architectures and real-world datasets.

---

### Meta-Review · Area_Chair_Juem · 2023-12-18

**Metareview:**

The paper considers gradient-based meta-learning MAML from bayesian inference persepective showing that vanilla MAML’s adaption can be seen as summing approximated Gaussian posterior distribution over the example without any reweighting. The authors Instead of using the same weighting, apply a Laplace approximation leading to different support examples having their own weight given by its own covariance matrix (which is the Hessian of the example’s negalive log-likelihood). The authors propose a method LAVA based on the above idea of weighting individual examples via the Hessians and aggregating them for the final adapted model parameters.

In the experiments the authors consider 3 few shot regression tasks and show that their approach approach reduces the variance of the parameter estimate and leads to improvements over the known baselines for this task.

The paper proposes a very reasonable approach and does a good job of demonstrating its potential. The writing is relatively clear however as some reviewers recommended there is a room for multiple improvements (see suggestions by reviewers). There are two primary weaknesses as highlighted by the reviewers of the paper as it stands

1. The paper's proposal is based on a computationally intensive procedure of Hessian computation. While the authors argue it can be parallelized however despite this being true computing Hessians will remain a significant bottleneck for the potential applicability.
2. The paper's experiments are on 3 tasks with relatively low inherent dimension. On the real world task of mini-Imagenet the paper shows no improvements (however shows comparable results). This further limits the potential applicabiilty of the method as the current set of experiments stand.

Given the weaknesses listed above I consider the paper to be slightly below the acceptance threshold and therefore my recommendation.

**Justification For Why Not Higher Score:**

The highlighted weaknesses of the paper in terms of computation efficiency and efficacy (over other methods) demonstrated on relatively low dimensional experiments.

**Justification For Why Not Lower Score:**

NA

---

### Decision · Program_Chairs · 2024-01-16

Reject